# Understanding Unmet Care Needs of Rural Older Adults with Chronic Health Conditions: A Qualitative Study

**DOI:** 10.3390/ijerph20043298

**Published:** 2023-02-13

**Authors:** Dennis Asante, Craig S. McLachlan, David Pickles, Vivian Isaac

**Affiliations:** 1College of Medicine & Public Health, Rural and Remote Health, Flinders University, Renmark, SA 5341, Australia; 2Health Vertical Centre for Healthy Futures, Torrens University, Sydney, NSW 2007, Australia; 3College of Nursing and Health Sciences, Flinders University, Renmark, SA 5341, Australia; 4School of Allied Health, Exercise and Sports Sciences/Faculty of Sciences and Health, Charles Sturt University, Albury, NSW 2640, Australia

**Keywords:** chronic disease, older adults, rural health services

## Abstract

Background: Rural populations experience poorer access to the necessary health services for chronic health conditions. Although studies of rural healthcare access continue to expand, most are based on quantitative data, yet normative views and lived experiences of rural adults might offer a better understanding of healthcare access and their specific unmet needs. This qualitative study sought the views of both rural-centric older people and healthcare professionals to understand health needs, barriers, and enablers of accessing health services, with a focus on chronic health condition(s). Methods: Between April and July 2022, separate in-depth interviews were conducted with 20 older people (≥60 years) in a rural South Australian community. Additionally, focus group interviews were conducted with 15 healthcare professionals involved in providing health services to older adults. Transcripts were coded using the NVivo software and data were thematically analysed. Results: Participants described a range of unmet care needs including chronic disease management, specialist care, psychological distress, and the need for formal care services. Four barriers to meeting care needs were identified: Workforce shortages, a lack of continuity of care, self-transportation, and long waiting times for appointments. Self-efficacy, social support, and positive provider attitudes emerged as crucial enabling factors of service use among rural ageing populations. Discussion: Older adults confront four broad ranges of unmet needs: Chronic disease management care, specialist care, psychological care, and formal care. There are potential facilitators, such as self-efficacy, provider positive attitudes, and social support, that could be leveraged to improve healthcare services access for older adults.

## 1. Introduction

The global population is increasingly ageing at an unprecedented rate [1,2]. Older adults (≥60 years) constituted 16.5% of Australia’s population in 2020, and this age cohort is estimated to reach 23% by 2066 [3]. The ageing process is associated with an elevated risk of multiple chronic health conditions [4,5], hence older populations have a higher need for chronic management services (e.g., palliative care, complex diabetes care, cancer management, and cardiovascular diseases care), specialist services (e.g., dental services, outpatient diagnostic, or therapeutic care), and services for mental and psychological distress that usually coexist with chronic physical conditions (e.g., symptoms of depression, anxiety, and emotional discomfort). Older adults with complex chronic disabilities may require assistance with personal hygiene, medication compliance, food preparation, and community engagement (formal care). The ageing–multimorbid health association and the functional decline have been found to become worse with geographical remoteness [4,6].

Rurally living older people are considered vulnerable for several reasons. Rural communities have limited access to medical specialist services. Depending on the size of the community, there may be limited health infrastructure, limiting acute and chronic medical care for the diagnosis and management of complex diseases [7]. Rural older Australians experience higher rates of complex multimorbid health conditions than their urban counterparts [8]. Furthermore, older adults in rural communities confront severe physical barriers, including the absence of public transport, long waiting times, and long distances to health centres, especially specialty appointments [9]. The extended waiting times for appointments, higher rates of provider turnover, and inadequate access to specialty care or chronic disease management services and support systems adversely impact the health and health outcomes of older people in rural environments [10].

Surprisingly, few research projects have investigated the healthcare needs and service provision challenges of rural older people in Australia [8,9,11]. For instance, exploring mental health service barriers for rural older people in Australia, Muir-Cochrane, O’Kane [9] interviewed 19 healthcare providers and reported two broad access barriers, namely, poor recognition of mental health problems and limited service availability. Similarly, Henderson and Dawson [10] found the fragmentation of governmental responsibility, funding regime, and centralization and standardization of service delivery as critical barriers in rural areas from the perspective of care providers. Mariño, Khan [11] studied patterns and factors of dental-care service use among older adults in rural Victoria. This cross-sectional study reported the existence of barriers such as cost, the length of the waiting list, and service availability to oral care. Indeed, most gerontological studies have adopted quantitative designs that may not be sufficient in capturing the lived experiences of rural older people and/or have underrepresented rural populations. Other international studies have demonstrated poorer access to health services among rural older people, which may be attributed to the higher burden of diseases reported in rural older adults [12,13]. Again, research examining need perceptions, barriers to care, and enablers of service use for complex chronicity in rural environments through the perspectives of both providers and older patients is limited. Accordingly, a more focused understanding of health needs and barriers to accessing healthcare in rural areas from the perspective of older people and health professionals is required for policy and practice.

It is not clearly understood how rural older adults perceive their access to healthcare, particularly during the period of COVID, and how this has affected barriers to needed services, their coping strategies, and their preferred policy directions to meeting their health needs. To comprehensively understand the perceptions of needs and barriers to care services, normative views from the consumer and supply sides would provide valuable insight. Such information may help in the cross-validation of common concerns or themes and offer an opportunity to identify any discrepancies in perceptions about care delivery. Moreover, the views of health service providers are crucial because the quality of services ultimately depends on them [13]. Therefore, we draw on qualitative interviews with rurally living older adults in South Australia (≥60 years) and local healthcare professionals to understand the health needs of rural older adults and barriers to accessing the needed healthcare. Hence, the aim of the study was to understand the perceptions of unmet health needs and barriers to healthcare across older adults and health service providers in rural South Australia. We also assessed their views on interventions believed to make services more accessible and responsive to the needs of older adults.

To determine this, the following specific questions served as a guide to the study:(a)What are the perceived unmet healthcare needs of community-dwelling older adults in rural South Australia?(b)What are the challenges to care access and facilitators of health-service utilization among rural older adults?

## 2. Theoretical Consideration for Healthcare Needs and Services Utilisation by Rural Older Adults

This study is guided by Andersen’s model of health services utilisation. This behavioural model of health service use was originally proposed in 1968 but has undergone stages of revisions [14,15]. The model allows analysis of contextual factors that interact to shape health-seeking behaviours. Central in this framework is that an individual’s use of health services balances across three functional domains including predisposing factors, enabling factors, and need factors [14,15]. Predisposing factors may include age, sex, race, occupation, and health beliefs [16]. Enabling factors are conditions that make health service resources obtainable to the individual and may comprise family and community resources, residential location, income, health literacy, and social capital [14,15]. Need factors are based on the principle that a person must perceive illness and a need for help before using health services [15]. Due to its robustness and flexibility for contextual analysis, several studies have applied Andersen’s model in health services research (e.g., [17,18,19,20]).

Employing Andersen’s model, Wandera and Kwagala [21] demonstrated the effectiveness of the model in predicting health services use among older adults in Uganda. The authors, however, specified that need factors such as the severity of diseases and mobility limitations were highly associated with health services use compared to the enabling and predisposing characteristics. Evashwick and Rowe [22] adopted the model to examine health services utilisation among older adults in the United States and found that need factors (e.g., diagnosed health conditions and perceived health status) were important predictors of physician visits, hospitalisations, and ambulatory care services use. Again, Weller and Minkovitz [23] demonstrated that enabling factors (e.g., the type of health insurance and financial status) were associated with health services use. Similar conclusions have been drawn by Rivara and Anderson [24] in a sample of American women who had experienced domestic violence. In this study, need factors were more highly associated with health services use than the other constructs of Andersen’s model.

Based on our research questions, we assume in this current study that need factors (complex chronic conditions) and the enabling/barriers (services availability, accessibility, social support, and provider attitudes) would shape health services uptake by rural older adults living with chronicity(s).

## 3. Research Design, Setting, and Methods

Qualitative studies can capture comprehensive information about beliefs, perceptions, and the lived experiences of a population of interest [25]. These qualitative data were gathered as part of a mixed-method study exploring ageing and co-morbid physical and mental healthcare needs in rural and remote South Australia (SA). SA is a regional State with a population of 1.8 million people of which approximately one-quarter of the residents live in nonurban areas. It is one of the fastest ageing states in Australia with a median age of 41 years, higher than the national median age of 38 years (Australian Bureau of Statistics, 2022). The median age in SA’s urban areas (39.3 years) is seven years younger than in its rural and remote areas (46.4 years) (Australian Bureau of Statistics, 2021).

In-depth interviews were conducted with older people, and three focus group discussions were organised with health service providers in rural communities. The decision to conduct individual interviews with older adults rather than a group setting was based on the ethical consideration of safeguarding participants’ health information (privacy issues). Since this study gathered information on chronic health conditions, it was not ethically approved for the disclosure of private health information to anyone except to only the research team members.

### 3.1. Research Team and Reflexivity

Interviews were conducted by the lead author (D.A.) who is a doctoral student, and the group discussions were moderated by the last author (V.I.), the project supervisor. Co-authors (C.S.M.) and (D.P.) provided critical clinical support during data analysis and interpretation. It is important to note that none of the study participants had prior relationships with the researchers.

### 3.2. Participants

We used a purposive sampling strategy to gather a range of opinions that were then used to recruit participants to determine differences and similarities in experiences, perceptions, and beliefs [26,27]. The participants included 20 older adults and 15 healthcare service providers. To recruit older adults, the first author sought permission from relevant organizational leaders where required and attended meetings and social gatherings (e.g., Rotary meetings, Zonta meetings, and Church services) with study flyers and information sheets to invite attendees to participate. Here, we approached community leaders with the project objectives, who then discussed the study with older people and directed those interested to the researchers for further briefing and recruitment. Eligible participants were 60 years or older with a self-reported chronic health condition(s) and who had used health service(s) in rural SA. Community-dwelling and cognitive capacity to participate were additional inclusion criteria.

A range of professionals was invited to participate in the study. The sample included generalists, nurses, mental health professionals, and social workers. We reached out to candidates via email with information sheets. Moreover, the first author (DA) visited hospitals and medical centres and distributed flyers and information sheets in care facilities. Two general medical practitioners, two mental health nurses, four general nurses, and seven social workers agreed to participate.

### 3.3. Data Collection

Older adults were interviewed by D.A. and V.I. in English between April and July 2022. Demographic data including age, health status, driving status (able to drive a car or not), and living arrangement (whether living alone or with relatives/friends) were taken. Subsequently, a semi-structured interview guide was followed with a series of open-ended questions about health needs, experiences and perceptions of health services access, challenges, facilitators including factors shaping their help-seeking and care, and opinions on how their care can be improved. Typically, the interviews lasted between 30 and 45 in the participants’ agreed-upon convenient environment.

Three focus group interviews were conducted with health professionals at different time periods. Two of the group discussions were conducted virtually via Teams software (Version 1.6.00.1381) with V.I. as the moderator. The third and final team discussion was moderated by D.A., and this was conducted face-to-face with two GPs in a medical centre. All focus discussions began with open-ended questions about participants’ work roles and responsibilities, professional backgrounds, and working experiences with providing care to older adults. Their views on the health needs of their older clients and access issues were sought. The group interviews lasted between 45 and 60 min. Of note, all interviews and/or discussions were digitally recorded and professionally transcribed for analysis.

### 3.4. Data Analysis

Two research team members (D.A and V.I) analysed the transcripts through a thematic framework [28,29,30,31]. A systematic multistage technique, namely, (a) familiarization, (b) identifying key concepts, (c) indexing, (d) charting and mapping, and (e) interpretation, was employed for the analysis [28,29,32].

During the familiarization stage, D.A and V.I reviewed the transcripts to grasp the content of the data. Using Andersen’s model of health services utilization as our theoretical framework, codes were developed through both deductive and inductive processes [33]. Predetermined codes based on the topic list for the interviews and review of the relevant theoretical constructs formed the basis of the development of initial codes (deductive approach). Here, need concepts such as diagnostic services, chronic management, and care for unspecific symptoms of depression and anxiety were coded. Self-transportation, inadequate professionals, access to specialists, etc. (barriers), provider behaviours, and help from relatives (facilitators) were coded. Additional concepts from the data were coded after iterative engagements (inductive approach). Investigator meetings were held during the coding process and, where required, the initial coding framework was modified through consensus.

In the indexing phase, a line-by-line examination of the data to match the codes was conducted by the researchers. Memos were used to annotate coders’ decisions and questions about the content of the transcripts and reflections on the analysis. Regular discussions were held on the annotations for the consistency of coding. The charting and mapping process involved organizing similar codes into themes and comparing emerging themes between older adults and health professionals. At this stage, we matched health professionals’ codes with identified themes from older adults’ transcripts when appropriate. Patterns and themes from both data sources (responses from older adults and health professionals) were compared for discrepancies and/or validation.

In the interpretation stage, major themes and typical quotes were identified to summarize the findings. We shared the results with participants for confirmation, and no disagreement emerged on the findings. NVivo (version less or 2020 edition) software was used to aid in the qualitative analysis.

Ethics approval: Ethics approval was granted by the Flinders University Human Research Ethics Committee (Project No: 4647). A written participant consent form assuring the anonymity of volunteers’ identites was signed by all participants. Participants were offered a shopping voucher valued at AUD$ 20 for their involvement in the study.

## 4. Results

### 4.1. Characteristics of Participants

We included 35 participants, with 20 older adults self-reporting chronic health condition(s) and 15 healthcare professionals. Among the older adults, the mean age was 63.65 (range 60–87) years, 11 were women, 5 could not drive, and only 3 were living alone. Most of the healthcare personnel in this study were social care workers (*n* = 7), mental health nurses (*n* = 2), general nurses (*n* = 4), and general medical practitioners (*n* = 2) (Table 1).

### 4.2. Summary of Major Themes

Based on the primary objective of this study, the findings from our data have been categorized into three prominent themes, namely, unmet health needs, access to healthcare services, and facilitators/enablers of health services utilization. Unmet health needs comprised chronic disease management, specialist care services, psychological distress, and formal caregiving. Access challenges included workforce shortage, continuity of care, transportation, and waiting time for an appointment. Facilitators involved health self-efficacy, social support, and positive attitude of service providers. Table 2 highlights the main themes, subdomains, and examples of corresponding quotes.

### 4.3. Unmet Needs

Older adults’ self-reported (chronic) health condition(s) including heart diseases, cancer, diabetes, sleep apnoea, asthma, glaucoma, depression, bipolar, arthritis, and osteoporosis. Chronic health conditions require a structured and well-coordinated management plan to prevent health decline and the onset of new conditions. Interestingly, our data highlighted many interrelated factors showing an association between reduced chronic disease management and a decline in physical activity and an increase in common mental health disorders. Both older people and health professionals placed an emphasis on the need for care.

### 4.4. Chronic Disease Management

Participants described factors including the lack of interprofessional coordination, unstructured chronic disease management plan, data transfer/sharing, and limited consulting time. Both patients and providers mentioned that the typical period of consulting that ranges from 15 mins to 20 mins is not enough for dealing with multimorbid conditions and screening for any new onset health condition or worsening of pre-existing conditions. A general medical practitioner indicated that … “*Often, almost everyone’s got a shopping list of six to eight things for 15 min, which means two minutes a problem with a quick hello as well. So, you’ve got less than two minutes a problem*” (GP1, male, 36–46 years). Older adults were more concerned about the handling of their health data records. They raised issues with the norm of carrying their health records around any time they visited new practitioners or facilities. One older patient in his 70s noted:

“*I do think record keeping on my health could be better. My personal record is up here, but it should be instantly available to all doctors anywhere……it simpler for a doctor to hone in on exactly what he’s trying to see based on what he’s thinking might be the problem with this patient.*”(Older participant 7, male)

Interprofessional coordination and a structured chronic disease management plan were primarily discussed by health professionals as the only effective and efficient way of managing chronic conditions of older people. There was a perception that a lack of coordination and a structured management plan affects the provision of needed services to a wider coverage of older adults. A senior generalist practitioner had this to share:

“*Everyone’s doing something different, no one’s working together with a clear plan of how the system or service is supposed to be working….. even just in between doctors, doctors will want to do their own sort of thing. The only way we can make it work and make it work really well is to make sure we’re super organised.*”(GP2, male, 25–35 years)

### 4.5. Specialist Care Services

Most of the older people reported that accessing specialty care meant waiting for specialists to come from urban Adelaide or making a trip to tertiary care centres located in Adelaide. Older people expressed being concerned about the long trips for some specialist care. Those unable to drive long distances rely on relatives to access specialty services. Specialist appointments were considered very difficult to secure by the study participants. Except for life-threatening emergencies, patients wait several weeks for appointments, a situation that health professionals feared could lead to the deterioration of chronic conditions, the onset of new conditions, and rapid functional decline due to delayed treatment. One participant stated:

“*Then in terms of visiting specialists, there’s often long waits to get into them for these more complicated conditions, if the specialist even comes up to the area as well and being three hours away from Adelaide, often people don’t want to travel, or they can’t afford to travel or they’re too sick to travel. If they can’t get into the specialist, we end up doing the care which is quite complex and complicated.*”(GP1, male, 36–46 years)

“*My heart condition has required operations and stents and all sorts of things which—that’s a specialist care field and you’ve got to go to Adelaide to access that. If I go to make an appointment, they say, oh well, it’s six weeks. In six weeks, the problem’s either a lot worse or it’s gone away.*”(64-year-old man, participant 5)

Older people expressed dissatisfaction with waiting too long to have access to physicians. One woman said, “*to see a specialist, like I see an eye specialist and it’s pretty hard. I book three or four months in advance for an annual appointment.*” (Participant 20, 73-year-old woman). The waiting time varied for different services (e.g., to see a GP vs. a specialist) and communities. One woman said, “*I think it’s disgraceful. I don’t know what the answer is but if you need to see a doctor, you should not have to wait four to six weeks to see the doctor of your choice necessarily I don’t think.*” (Participant16, 71-year-old woman).

Even on the day of the appointment, patients are sometimes made to wait several hours before they are able to see a doctor, a situation that is, according to them, very stressful to bear given their health conditions and age. Healthcare providers agreed that obtaining access to doctors, particularly specialists, is sometimes difficult.

### 4.6. Psychological Distress

More than half of the older adults reported two or more health conditions, and most of these people discussed psychological distress associated with their physical condition. There is a perception among older people that psychological distress and other distress that might be coexisting with their chronic conditions do not capture the attention of doctors as much as their physical conditions. A 75-year-old cancer patient shared her experience of untreated depressive symptoms that coexisted with her physical condition “*… Interestingly, I think there was a depression associated with that that wasn’t treated at the time. Now, I’ve got ongoing management…*” (Participant 11, male). GPs described their awareness of patient distresses normally coexisting with multimorbid physical health conditions … “*We see a lot more chronic depression and anxiety around poor health conditions.*” (GP2, male, 25–35 years). However, the time limitation does not allow for screening and treatment unless there is a manifestation of severe symptoms:

“*Yeah, we just don’t have the time to do a DASS-21 or a DASS-42 or something like that…. But for a standard GP consult in 15 min, there’s normally so many medical conditions, acute and chronic. Even if it’s just a chronic disease consultation, that person always brings in some acute problems as well.*”(GP1)

Healthcare providers also discussed the complexities in the health system as a source of mental stress to older adults. As a mental health nurse put it: “*it’s not just the access, but it’s that ongoing navigation of our health system. you’ve got most*—*a lot of people have more than one chronic illness. They’ll have two or three. They’ll have a combo of things that trying to get all those specialists or people working and the client doesn’t know where they’re at with all their medications and things like that*” (Participant 22_Mental health nurse, male, 36–46 years)

Some older adults did not consider symptoms of depression and emotional distress as medical conditions requiring formal assistance but rather as normal symptoms of ageing.

### 4.7. Formal Caregiving

Older adults discussed the need for support in carrying out activities of daily living. For instance, participants frequently mentioned needing domestic assistance (e.g., cleaning, cooking, and gardening). They also described the need for help to do grocery shopping and access certain public places. Older adults indicated a preference for receiving help from formal support networks or organizations. Health professionals, particularly social workers and nurses, added personal care (bathing, laundry, and hair grooming), transport, and assistance with medication. As one social worker put it:

“*We need to put in personal care, because when we—people need to shower, they need to look after their skin integrity. Older people become incontinent; how do we manage that? So somewhere in there, I’d like to see personal care and transport put in.*”(Participant 26, female, Soscial worker, 25–35 years)

### 4.8. Challenges to Healthcare Access

From our discussions with the participants, four categories of barriers to health services access emerged: Workforce shortages, continuity of care, transportation, and difficulty scheduling appointments/long waiting times. Of note, patients described their challenges accessing health services and difficulties they believed their peers also faced.

### 4.9. Workforce Shortages

Participants were focused on the limited number of healthcare practitioners in their localities and linked this phenomenon to unbearable workload (on providers) and limited consulting time and/or limited healthcare supply. Discussion about the inadequate number of health professionals included difficulty attracting GPs, specialists, nurses, and other care providers to rural communities. One man in his 80s said, “*they are obviously overworked and tired. Not enough of them, too many of us. Too many patients, not enough doctors*”.

Many older adults find the high healthcare professional turnover rate in their communities very problematic. However, they believed limited opportunities in rural locales are to blame for the situation, as one woman indicated, “*Once they do their thing up here, next minute, they’re gone. But then I suppose, too, a lot of them, their*—*once their children get to a certain age, the better schools are down that way, where the kids [that] go to uni, so they tend to move*.” (Participant 12). In discussing staff shortages, a social worker had this to say, “*I think it comes down a lot to, once again, there’s a staff shortage. We don’t have enough support workers. There are very few allied health disciplines within Port Pirie*.” (Participant 27).

### 4.10. Continuity of Care

The lack of continuity of care emerged as one of the critical barriers to required healthcare. Two main issues arose, namely, irregular or unstructured appointments and a lack of regular doctors or widespread provider turnover. Older people wanted to see a particular doctor for a longer period to build trust and for the doctor to know them and their conditions well. This, they believed, will ensure the continuity of patient care, minimise medical errors, and promote the efficient use of limited consulting time. One man shared his frustration, “*What I want is I want one doctor*—*just one*—*who doesn’t have to be the world’s best doctor, but I want them to know about me. I don’t want every time I go to the doctor to have to explain this and this and this and they get on the computer and they*—*oh yes, I see.*” (Participant 19, Man in his 60s). In the words of a 70-year-old woman, not being able to see her regular doctor has resulted in a medical error that she must live with. An excerpt from her account is as follows:

“*Very hard to see the GP that you want to see. I had to see a registrar when my voice went strange, and he told me it was laryngitis, and it was lung cancer. So, I was a bit annoyed, but we all make mistakes, don’t we?……but it’s nice to see your own doctor who knows you. Because he knows me, he would have known that I’d lost a bit of weight and he knew as soon as he saw me in the supermarket that something was wrong. So, if I’d seen him, maybe things would have been a bit different, but who knows?*”(Participant 9)

There was a perception among older adults that new providers normally do not have enough time to abreast themselves of the conditions of patients … *Yeah, like the man with my voice. Maybe if he’d read a bit more of my history, maybe he would have been a little bit more thorough.*

On the part of health professionals, continuity of care promotes physical and mental wellbeing outcomes. As a generalist put it *… if you’ve got good continuity of care, you can normally pick up when someone’s mental health has dropped off. Even if we see them four times a year, it’s enough.* (GP1).

### 4.11. Transportation

Issues of transportation were featured significantly in the interviews. Primarily, transportation barriers included limited public buses, traveling out of town for specialty appointments, the inability to drive longer distances, parking problems in the city, and limited medical transportation programs. Many older adults reported they travel long distances to urban areas for health services that went unmet in their communities, particularly for specialists’ consultations. One man commented, “*My heart condition has required operations and stents and all sorts of things which*—*that’s a specialist care field and you’ve got to go to Adelaide to access that.*”(participant 5). One woman indicated that without assistance from friends and relatives to drive her, she may not be able to attend her appointments as she did not feel confident to drive, especially after medication *… I possibly could but I probably don’t feel confident to drive after I’ve had the injection in the eye.*” (Participant 8, 72-year-old woman). Similar sentiments were shared by the health professionals. They described transportation as one of the frequent reasons older people cited for missed appointments.

### 4.12. Facilitators of Health Services Utilization

Despite the challenges of accessing health services, most older adults we interviewed described three broad factors including self-efficacy, social support, and positive attitudes toward health services providers that help them to navigate the healthcare system to access their healthcare needs.

### 4.13. Health Self-Efficacy

Older adults who described themselves as more assertive, hopeful, and aware of the benefits of healthy behaviours and asked questions about their conditions and treatment options reported that they always feel encouraged to seek help whenever needed. As a 72-year-old man said when asked to share factors that facilitate access to needed services:

“*I’ve been involved in a number of health programs, reaching out for the community. Stress management, stop smoking programs. One time, weight loss and because of my awareness of—it’s the importance of maintaining good health, I think that’s sort of helpful in my situation. It keeps me focused that if there’s something that I feel is not right, I like to get it checked out.*”(Participant 12)

Another woman shared, “ *I’ve never been frightened to actually ask something or get them to explain something if there’s*—*yep.*” (Participant 1, 65-year-old woman).

Healthcare providers reported that older people who are more proactive and confident in asking questions tended to make regular appointments and follow-ups to improve their conditions. As a nurse discussed, “ *The people who are proactive often get a better outcome because they’re prepared to take responsibility for their own needs, and they’re prepared to ask for help. Their prepared to find out what needs to be done*.” (Participant 27).

### 4.14. Social Support

Getting help from friends and relatives to attend out-of-town appointments including specialty care appointments in Adelaide, encouragement, and financial support from relatives, according to older adults, were very crucial in meeting their healthcare needs. As exemplified in the words of a woman in her 70s, “*I haven’t got a car, so they’re [her children] driving around everywhere. I just tell them straight out, I say you drive. So that’s it, they take me there.*” (Participant 13). On the other hand, older people with an inadequate social support network sometimes delayed their treatment or lost out on their medications. A 65-year-old woman commented, “*sometimes if you go down to Adelaide and you need to stay there, those are just extra costs and that - and then also getting medication is an extra cost. Sometimes, I don’t even get my medication because I can’t afford it*.” When asked whether she could receive support from friends and relatives, she responded, “*No, because they’re in the same boat*.” (Participant 4).

### 4.15. Positive Attitudes of Service Providers

Even though older people discussed various struggles in accessing healthcare services, most spoke highly of their healthcare providers. Their primary concern had to do with the lack of a long-term, trustworthy relationship due to the regular change in doctors. However, older adults generally reported satisfaction with their providers. In discussing their experiences with health services providers, older people frequently used phrases such as “*they’re accommodating*”, “*they’re respectful*”, “*they’re polite*”, “*they’re quite good*”, *they’re doing their best*”, and “*quite pleased*”. The professional conduct and positive behavioural disposition of care providers, according to some older people, encourage them to use the needed services. In response to the question “what is your experience with the healthcare system and providers?” one man submitted:

“*I think they’re doing a—I value the doctors who own the practice, I value their experience. If I can, I’ll see them.*”(Participant 1, 66-year-old man)

Healthcare providers sometimes went beyond their responsibilities to help older patients to prevent conditions from worsening. They noted that such decisions were normally made based on risk assessment and in strict consideration of professional ethics. A social worker described a typical case: “*sometimes, we*—*depending on risk, we might need to transport* [patients] *ourselves …. Again, not really our business, but we can’t sort of ignore that so we then reopen another referral and help them navigate that whole process*” (Participant 23, Social worker).

### 4.16. Suggested Interventions to Address Access Barriers

In our interviews with the participants, we sought their opinions on interventions or programs they believed can help reduce the barriers to health services delivery and access. Although both healthcare providers and older people proffered solutions that targeted workforce shortages more broadly including recruiting and training more locals, attracting professionals from abroad, and instituting incentive packages for rural practice, healthcare providers further suggested the need for structural changes in the healthcare system. Further discussions on the suggested interventions are being developed as part of the broader project objectives into a manuscript. However, below are excerpts of the suggestions made by the participants:

“*We’ve got to attract them to the country, and we’ve got to retain them in the country…If you’re a doctor in the country, you’ve got an additional cost associated with educating your kids, if you want them to be privately educated, and accessing additional training is all much more difficult if you’re a doctor in the country. So, I think the answer is to recruit more doctors to the country*”(Participant 13)

“*We’ve got to consider other options……why can’t we attract doctors from abroad? we’ve had different—doctors from different countries in our local town and we’ve found them to be just as good as the Australian doctors*”(Participant 7)

“*I think yeah for a medical clinic or any medical service to be successful in providing good chronic disease care, both physical and mental health issues, the whole clinic or the whole service needs to be working together and on the same page. There’s no point… We’ve seen it. There’s no point where the doctors will be doing one thing, the receptionist will be doing another thing, and the nurses will be doing one thing. Everyone’s doing something different, no one’s working together with a clear plan of how the system or service is supposed to be working. So, certainly we inherited a very broken system*”(GP 2)

## 5. Discussion

We conducted a qualitative study with older adults with chronic health conditions and healthcare providers in rural areas to understand their perceptions of needs. The data highlighted four interrelated unmet care needs: (a) Chronic disease management, (b) specialist care, (c) psychological distress, and (d) formal caregiving. Participants discussed four major barriers to health services including workforce shortages, the continuity of care, transportation, and waiting times for appointments. These challenges are partly mitigated through health-related self-efficacy, social support, and the provider’s positive attitudes. Andersen’s health services utilization framework [14] provides useful insight into understanding the access to and use of needed care services (see below). Our study provides valuable insight into how service provision to older rural people could be improved to meet their expected needs, particularly in those with complex diseases or two or more chronic conditions.

Both respondent groups discussed suggestions believed to improve health services delivery and access for older adults. Overall, the findings highlight the critical gaps and opportunities for a geriatric care model for rural older adults with chronic health conditions.

This work builds on prior literature on older people’s experiences with rural healthcare systems in accessing health services. As reported in earlier studies [9,26,34,35], this study confirms that older adults with a chronic health condition(s) require a range of healthcare services that remain to be addressed in rural and remote areas. It has been reported that “rural areas have significant gaps in the continuum of care since community-based long-term care services are often unavailable” [36]. The study highlighted the need for improved chronic disease management and addressing of psychological distress and better access to specialty care for complex multimorbid conditions that older individuals in rural environments are prone to experience [4,6,37]. This finding is in keeping with a similar Australian qualitative study, which reported unmet specialty care needs of rural and remote older adults [34]. However, our data also show additional domains of service gaps that are critical to the health and wellbeing of older people. While this study recruited both older adults and providers to understand their perspectives of health needs and barriers to accessing services, the earlier study [34] only reported data on provider perspectives; a possible reason for the discrepancy in the findings.

The unmet care needs of rurally living older adults have been a longstanding public health concern in many parts of the globe, and this phenomenon is attributed to exacerbating barriers to healthcare services in rural locales [26,35,36,38,39,40]. Access barriers such as workforce shortage and provider turnover are common challenges in rural health systems [41]. Workforce shortage and provider turnover are related to the problems of continuity of care and long waiting times for appointments in this study. Therefore, attracting and maintaining adequate rural health practitioners will significantly reduce the long wait times and lack of continuity of care. Older adults build trust and are more confident with providers they have seen for a reasonable period [26]. Shorter periods of employment for care providers and long wait times may impact older adults (across multiple domains including physical health and psychological wellbeing) who are likely to require frequent and complex health services as they age [26,36].

An important care need identified by participants that has not been previously discussed in the population studied was formal care. Participants reported that chronic health and formal care needs were addressed as separate issues for older adults, with a limited understanding of how these are interrelated. Available studies on the unmet care needs of rural older adults have reported mental health services [9,34], chronic health management services [42], and specialty diagnostic services [43]. According to the study participants, especially the care providers, complex chronic conditions may impact older people’s ability to effectively adhere to routine medications, personal hygiene activities, and food preparation and proper dieting. This may, in turn, impact the health and well-being of older people. This suggests that holistic care is likely to be better for rural older individuals with complex health conditions.

Transportation was also identified as a critical barrier to accessing the healthcare needs of rural older adults with chronicity. Accessing care that is not available in local communities, especially specialty care, normally required long-distance travel to urban areas. Some older adults discussed their inability to cope with long-distance driving and, hence, made them dependent on relatives to access such services when there is no public transport. A similar Australian study [9] and another synthesis of primary evidence on barriers to health services [40,44] reported transport barriers to health services use among rural older adults. Moreover, qualitative research on the barriers to accessing specialist care among older people with chronic obstructive pulmonary disease in rural New Zealand indicated, among other challenges, transport barriers [43]. It is important to note that navigating these barriers is associated with significant levels of psychological distress and anxiety [6,43].

For older people to access the required health services in a timely manner, self-efficacy, social support, and positive attitudes of healthcare staff are crucial. Self-efficacy has been shown in the general population to be associated with positive health behaviours [45,46]. The current study explicitly demonstrates how self-efficacy facilitates health services use among rural older adults with complex chronicity. Self-efficacy elements were associated with older adults’ confidence and inner drive and may represent self-cognitions with respect to a desire to achieve their best possible health and wellbeing [46]. Self-efficacy is very important in complex chronic disease management and is associated with the appropriate use of health services to improve health outcomes in multimorbid patients [45]. This observation implies that higher self-efficacy may result in enhanced wellbeing for chronically ill older adults. Hence, an awareness of the self-efficacy levels of older patients may assist providers to identify those in need of enhanced self-management support [45]. Further, as anticipated [9,26], social support and positive attitudes of care professionals were crucial facilitators of accessing healthcare among older adults. Bardach and Tarasenko [47] emphasized the importance of social support domains for rural older adults with significant medical needs.

Our participants shared their opinions on how the perceived barriers to healthcare could be addressed. Both older adults and care providers concurred with the need to address the inadequate workforce to reduce the workload of current practitioners and improve waiting time for appointments. Suggested strategies, however, differed slightly between providers and older adults. While providers were in favour of structural adjustments to attract more providers and ensure efficient use of care resources, patients tied workforce shortages to limited economic gains and suggested financial incentive packages to attract more providers to rural areas.

### 5.1. Theoretical Perspective

As explained earlier, Andersen’s behavioural model of healthcare services use is a widely applied framework to analyse health-seeking behaviours. The framework assumes that health service use is a function of three constructs, namely, predisposing characteristics (e.g., age, health beliefs, and gender), need variables (perception of illness or diagnosed health conditions), and enabling factors (e.g., wealth, availability of services, and support networks) [14,15]. Our findings contribute to the understanding of the need and enabling constructs of this model. We have highlighted common unmet needs of rural older adults with complex chronic conditions, corroborating the need variable of Andersen’s model (Figure 1). Regarding the enabling factors, this study demonstrates that perceived positive attitudes of providers are an enabling resource for timely health services uptake by rural older adults. Self-efficacy has been shown to facilitate the appropriate use of needed health services [48]. Moreover, social support is an enabling resource for health service use in a rural context. On the other hand, contextual factors such as transportation problems, long waiting times, workforce shortages, and the lack of continuity of care serve as critical barriers to accessing the required healthcare services.

### 5.2. Limitations

The findings of this study should be interpreted considering the following potential limitations. First, the study sample of older adults were from rural SA. As such, the generalisation of this study’s results to different states would need further replication studies, such as, in different Australian regional areas. Second, the views shared by participants, especially healthcare workers, may be influenced by patient privacy issues and social desirability bias. Again, we did not stratify the views of healthcare professionals in the focus group interview. It is plausible that if we probed participants separately, differences in perceptions or contradictions may occur. However, the aim of the study was to explore unmet healthcare needs, barriers, and enablers in accessing health services from the perspectives of demand and supply sides in a composite manner. In this regard, the study has contributed to knowledge in several ways. Theoretically, the study shows the applicability of Andersen’s model in qualitatively investigating the lived experiences of rural older adults in accessing healthcare. Through this, how various factors interact to shape health-seeking behaviour among older people in rural environments is demonstrated. The results also augment the literature on older adults with complex healthcare needs in rural environments. Again, this study is among the first to sample the voices of rural older adults and their healthcare providers in evaluating needs, barriers, and enablers of health services use. Hence, more qualitative studies will be necessary to explore this and other relevant details to contribute information and potentially shape policy by providing a forum in which older people and rural health practitioners can voice unique issues related to the rural healthcare system.

## 6. Conclusions

We explored the perspectives of older adults and healthcare professionals on health needs, barriers, and enablers of accessing health services among rural older adults with a chronic health condition(s). Older adults confront four broad unmet needs: Chronic disease management care, specialist care, psychological care, and formal care. Many access barriers have contributed to the unmet care needs; however, there are potential facilitators that could be leveraged to improve healthcare services access for older adults. The findings reinforce the need for governments and other stakeholders to act effectively to think of ways to improve care services in rural areas. Particularly, the existing system of formal care services provided in areas of personal hygiene, daily activities, and engagement in community and social activities that may be strengthened. Importantly, the provision of geriatric care services in rural settings needs to be reassessed and efforts should be made to train more professionals through scholarships and other appropriate incentive packages to work in rural care facilities. The study also provides new insight into the care needs and facilitators of services use. Particularly the impact of health self-efficacy on health services use in late life and formal care needs, which has not been previously reported. The awareness of self-efficacy levels of older patients may assist providers in identifying those in need of enhanced self-management support. Overall, the findings of this study provide valuable data on perceived unmet healthcare needs and barriers to and facilitators of health services use (answering the research questions posed) and highlight the need for comprehensive stakeholder engagements, particularly older adults, in an effort to address their care needs in rural environments.

## Figures and Tables

**Figure 1 ijerph-20-03298-f001:**
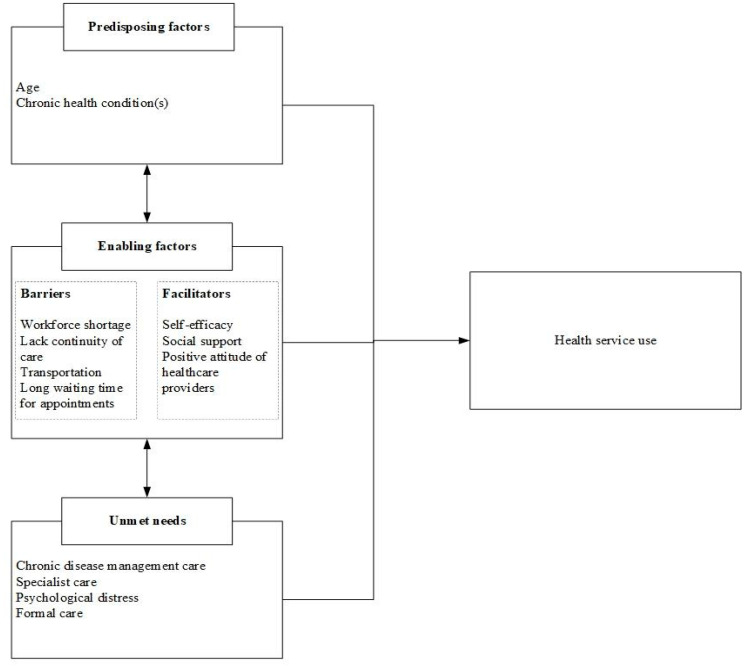
Unmet healthcare needs and factors of older adults’ health services use.

**Table 1 ijerph-20-03298-t001:** Participant characteristics.

Characteristics		
	Older Adults (N = 20)	Healthcare Providers (N = 15)
Mean age (years)	63.65	
Age range	60–87	25–35 = 9, 36–46 = 4, 47–57 = 2
Sex	Male = 9	Male = 5
	Female = 11	Female = 10
Professional background	-	General practitioners = 2
	-	Mental health Nurses = 2
		Social Workers = 7
		General Nurses = 4

**Table 2 ijerph-20-03298-t002:** Unmet health needs, access challenges, and facilitators to health services utilization.

Themes	Patient Quote	Provider Quote
Unmet needs		
Chronic disease management	“I have type 2 diabetes, lupus and stage 4 lung and brain cancer…..Every three weeks for four treatments and then after that it’s every three weeks for 24 treatments, which I’m trying to have up here in Berri, I hope.” (70-year-old woman) Because the dollar return per minute on a short consult is higher than the dollar return per minute on a long consult. So, they want more of the short consults because it raises more money, and I thought, what’s the object of the practice of medicine here? Is it to raise more money, hence we want shorter appointments, or is it to provide a level of care that’s going to meet the need of the patient? (75-year-old man)	“The other thing with chronic conditions, people with chronic conditions need, I think they really need the doctor to go the extra mile to do all their preventative care unrelated to their other conditions as well.” (GP) “Often, almost everyone’s got a shopping list of six to eight things for 15 min, which means two minutes a problem with a quick hello as well. So, you’ve got less than two minutes a problem.” (GP) “I think yeah for a medical clinic or any medical service to be successful in providing good chronic disease care, both physical and mental health issues, the whole clinic or the whole service needs to be working together and on the same page. There’s no point… We’ve seen it. There’s no point where the doctors will be doing one thing, the receptionist will be doing another thing, and the nurses will be doing one thing”
Specialist care services	“The obvious other one is access to the level of medical care that you need. My heart condition has required operations and stents and all sorts of things which—that’s a specialist care field and you’ve got to go to Adelaide to access that. So, there’s something about the level of medical care not being accessible locally but I don’t think we’ve got the population to justify everything for everyone. We do need to understand that’s one of the costs of living in the country.” (64-year-old man) “It’s seeing specialists, because you’ve got to travel down to Adelaide. That’s over 200—about 200 and something kilometres, and organising all of that.” (65-year-old woman)	“if we can keep the physical and emotional health quite good with the lack of specialists and lack of other health care up here, it’s a huge win for people.” (GP) “I’ve had a recent situation with someone in a regional town. They needed a significant amount of support and the doctor basically said well you need to put your wife into aged care, and you need to move to the city so you can get the support that you need.” (social worker)
Psychological distress	“I think there was a depression associated with that that wasn’t treated at the time. Now, I’ve got ongoing management of those, which is both a combination of local and Adelaide-based treatments.” (Man in his 60 s). “I used to worry so much about things but that made me sick so you learn how to just—I don’t know, just—you’ve got to cope, somehow you cope. I suppose at 65, you don’t worry about the little stuff so much. Because the main thing at this age is your health.” (Woman in her 60 s) “I haven’t considered that the anxiety has been long lived. It just might be because of a situation that’s arisen. Then we can resolve it within a very short period of time.” (70-year-old man)	“Yeah, we just don’t have the time to do a DASS-21 or a DASS-42 or something like that. The nurses in some of the health assessments will do a geriatric depression scale, or mini mental state exam or something like that, because they get given 45 min. But for a standard GP consult in 15 min, there’s normally so many medical conditions, acute and chronic. Even if it’s just a chronic disease consultation, that person always brings in some acute problems as well.” (GP) “time is a huge one, usually for proper consultant for mental health, you often need at least 20, 30 min, and that’s doing it fairly superficially, and not doing a great deal more.” (GP)
Formal caregiving	“There’s just things around the home that I’m looking at whether I can get support with those. That’s with things like the bathroom cleaning and general cleaning………Now, with something like cooking tea, my back can be tight by the time I finish cooking tea, and yeah, so I’m just seeing if I qualify for anything. Otherwise, we’ll get somebody in to just clean periodically and do drop-ins once a month” (Man in his 70 s).	“I also see that somewhere, we need to put in personal care, because when we—people need to shower, they need to look after their skin integrity. Older people become incontinent; how do we manage that?” (Social worker) “Talking from a perspective of community services under Commonwealth Home Support Program, the basic care needs really commence with domestic assistance and social support, respite for the carer, shopping assistance for those very basic needs which become the first they become evident when people are wanting to stay at home and their physical health, whether it be through chronic condition or issues of ageing, prevents them from doing those activities themselves for the long-term.” (Community health nurse)
Access challenges		
Workforce shortages	“Doctors—the hours of our medical staff—our healthcare providers generally, the hours that they’re working are ridiculous. I don’t understand how they can operate safely. If you put that degree of work pressure onto other professions, they would fold. If we made our air traffic controllers work like our doctors, we’d have planes crashing often. So, I just think we expect too much of our doctors and why? Because we haven’t got enough of them. We need more doctors.” (Woman in her 70 s). “There is certainly not enough doctors in this place. That’s the simple truth of it.” (80-year-old woman).	“But then just one of the big ones it’s just access, because there’s just not enough general practitioners to cope with the load. There’s not enough psychiatrists, not enough people in mental health teams, there’s not enough psychologists, not enough counsellors, like access is difficult.” (GP) “I think it comes down a lot to, once again, there’s a staff shortage. We don’t have enough support workers. Certainly, when I listen to the support workers and they’ll talk about their workload. Some of them now are only finishing at six o’clock at night, because that’s what they need to do because there’s not enough workers to provide the services that are required” (nurse)
Continuity of care	“I’ve only been here for years myself, so I haven’t got what I would call a regular doctor.” (73-year old man) “What I want is I want one doctor—just one—who doesn’t have to be the world’s best doctor but I want them to know about me. I don’t want every time I go to the doctor to have to explain this and this and this and they get on the computer and they—oh yes, I see.” (Man in his 60 s) “It’s the change of doctors you have all the time. It’s all right if you have the same doctor, because when you have different doctors, I don’t think they should read your notes.” (65-year-old woman).	“Well, I think since we’ve started doing the chronic disease management a lot better and a lot more structured, you get better. You can actually stave off a lot of chronic diseases worsening by just dealing with them regularly, which is just been a major change that we’ve done, going from ad hoc appointments to some really structured follow up the physical side are actually seeing a lot of better health outcomes for our patients.” (GP) “So, yeah, I’d agree that if you’re doing good, if you’ve got good continuity of care, you can normally pick up when someone’s mental health has dropped off. Even if we see them four times a year, it’s enough.”
Transportation	“I’m not allowed to drive at the moment due to the brain cancers. I only drive locally anyway, so if I need to go to Berri or further, I need to get friends to take me.” (Woman in her 70 s) I possibly could but I probably don’t feel confident to drive after I’ve had the injection in the eye.	“We have a lot of people with significant issues that just can’t leave home, either physically or emotionally, which is huge, up here as well, because you’ve been three hours away from Adelaide. Where if your health is terrible, you’re a long way from a tertiary centre and you just can’t access certain things.” (GP) “As people get older, they’re not confident driving long distances, they often have to go to Adelaide for medical appointments, or to Port Pirie, and transport becomes a big thing. They can’t get onto a bus, there’s no bus services available.” (Social worker)
Waiting time for appointment	“ if you need to go see a podiatrist or you need to go see a dietitian, the waiting lists are long, so long. Sometimes I’ve been—you have specialists that come up to the Riverland and a lot of the times, they’ll recommend you; it’s easier to—and quicker if you just go to Adelaide to see them, rather than waiting to get something up here.” (Woman in her 70 s). “You have to wait a fair while to see a specialist, like I see an eye specialist and it’s pretty hard. I book three or four months in advance for an annual appointment.” (73-year-old woman)	“The role that I’m in now, which is supporting people with ongoing chronic conditions, sometimes they might have five chronic conditions at the same time, including rare genetic disorders. If people had assistance earlier, the outcomes could be a lot better than what I’m seeing every day in my work. Just the massive barriers for people actually getting access to the services that they do need.” (Nurse) “Then in terms of visiting specialists, there’s often long waits to get into them for these more complicated conditions.” (GP)
Facilitators		
Self-efficacy	“Well the physical things—I do lots of physical exercise and try and keep myself very fit. I do Pilates, I walk, I do aerobics—so I’ve been trying to fix my body myself, yeah. When I need to see a doctor, I go for appointment.” (61-year-old woman). I’ve been involved in a number of health programs, reaching out for the community. Stress management, stop smoking programs. One time, weight loss and because of my awareness of—it’s the importance of maintaining good health, I think that’s sort of helpful in my situation. It keeps me focussed that if there’s something that I feel is not right, I like to get it checked out. (72-year-old man)	“I’m relatively new into the health space, I’ve only been here 12 months, and I still don’t feel confident in navigating certain parts of it. I’m coming from an educated background, with the ability to advocate for myself if I need it. A lot of the clients that I see, actually don’t have that.” (Social worker) “The people who are proactive often get a better outcome because they’re prepared to take responsibility for their own needs, and they’re prepared to ask for help. Their prepared to find out what needs to be done.” (Nurse)
Social support	“They are accessible but because of the glaucoma I’m having injections in one eye and I have a brother in Berri who will come and get me and take me home again. Also, a brother-in-law who will come and get me and take me home.” (78-year-old woman)	“Family support is also wonderful, but a lot of people in Port Pirie—and Mel will attest to this—they go into hospital, they don’t even have someone who can feed their cat or go and get them a bag of clothing because they just don’t have a support network.” (Social worker)
Positive service provider attitude	“I’ve been quite happy with the reception of the nurses and the reception staff because they’re very polite [here]. Even at Berri they’re quite good too, the specialist.” (Woman in her 70 s)	“sometimes, we—depending on risk, we might need to transport ourselves. So we don’t promote that, but we talk about it as a team and if we need to do it, we will do it….. Again, not really our business, but we can’t sort of ignore that so we then reopen another referral and help them navigate that whole process” (Social worker)

## Data Availability

This study’s data cannot be shared due to ethical (IRB) considerations. Again, the authors are still conducting studies with the data for future publications.

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
