# Peer review of "Understanding Unmet Care Needs of Rural Older Adults with Chronic Health Conditions: A Qualitative Study"

_ijerph, 2023, doi:10.3390/ijerph20043298_

Round 1

Reviewer 1 Report

Thanks for the opportunity to review this work “Understanding unmet care needs of rural older adults with 2 chronic health conditions: A qualitative study”. The issues discussed in the work are great and contribute greatly to the scant information in the field especially in rural Australia. Looking at the fast-growing rate of older adults’ population globally and Australia in particular, a better understanding of  unmet care needs among older adults is key in helping to achieve the UN Decade of Healthy Ageing (2020-2030). The authors have shown a great understanding of the subject under investigation. I have few comments to sharpen the paper.

1.      For a qualitative study, you need to provide information on research team and reflexivity.

2.      To assist international readers, kindly provide information on the study setting.  

3.      Comment on how you ensured trustworthiness/rigor such as dependability, credibility, confirmability, and transferability in your study.

4.      Under the ethics, providing only institutional approval is not enough. You need to include information on consent, privacy, and confidentiality

5.      If you are commenting on the Andersen’s health services utilization framework (Andersen, 1995) as your theoretical foundation, then why don’t you let readers know at the background/introduction section that you are using this framework to guide your work. It is not appropriate to just jump the theory in the discussion. Don’t take readers by surprise! I suggest you provide a paragraph information on the framework in your introduction. Just 5-6 lines in the discussion should do.

6.      Include the strength (s) of the study.

7.      Remember, you have raised two questions in the introduction. As it stands now, these two questions are still begging for answers. Why because, we are still not clear whether your findings have been able to address them. Thus, you need to tie your conclusion to these two questions by telling readers whether your findings have provided enough information to answer the research questions. Readers will be happy to see that. You should not expect readers to keep wondering.  

Reviewer 2 Report

General comments

===============

The paper "Understanding unmet care needs of rural older adults with chronic health conditions: A qualitative study "aims at understanding "perceptions of unmet health needs and barriers to healthcare across older adults and health services providers in rural South Australia." (see p. 2, line 71-72)

In terms of methodology, the authors have applied a qualitative approach by using focus group interviews with 15 healthcare professionals involved in providing health services to older patients and 20 in-depth interviews with adults who were older than 60 years. Unfortunately, the manuscript did not reveal the interview guideline and the questions raised. Consequently, the data analyses are difficult to follow and not completely understood. The data were collected in the timeframe between April and July 2022.

The manuscript, in general, is easy to understand and well-written in style as it is the joint endeavor of a team of native English speakers. However, at least some sentences should be read through again diligently as there seem to be some minor grammatical errors in the text (at least from my non-English native-speaking view).

The overarching theory included in the study seems to be Andersen's model of health services utilization, as mentioned in the Data Analysis section (see lines 137-138). However, the theoretical framework has not been explained in the Introduction section in detail and, thus, seems to serve as some fig-leaf function in the present manuscript form. Although it is mentioned that some categories were deduced out of the theoretical framework, no specific details are given in this respect. Thus it remains unclear which categories were drawn out of the theory.

The general topic is very interesting, bearing the aging population more or less worldwide in mind. The title sounds very promising and can satisfy raised expectations in its present form.

I have read the manuscript with great interest. Nevertheless, some weaknesses should be addressed regarding the paper, which must be resolved before it can be seriously considered for publication in the target journal. I want to elaborate on the weaknesses of the present manuscript in detail as follows. With a manageable effort, it should very easily be possible to publish the manuscript soon.

Please regard the following points as constructive criticism.

Specific comments

===============

Major comments

-----------------------

11. The literature basis of the present manuscript under consideration is sufficient. Nevertheless, I would invite the authors to underline the golden thread of the Introduction and rethink its structure. I would expand the summary of what has been known so far on barriers and facilitators of health services utilization for the elderly and identify a clear research gap based on existing literature. This would make the core contribution of the manuscript more visible. Thus, I would invite the authors to expand the literature review in the Introduction and systematically explain which barriers and facilitators have been identified from the patients and the perspective of healthcare providers in the literature so far. Also, a clear reasoning for the qualitative approach and focusing on two perspectives, i.e., the patients' and the health providers' perspectives, should be included. Arguing that health service providers are crucial because service quality depends on them is a weak argument, at least in my view. Opting for leading group discussions and interviews with these two target groups should be justified.

2. Additionally, I would cordially invite the authors to explain Andersen's model of health services utilization much more in detail and to explain which categories were deductively used for the data analysis. In its present form, Andersen's model of healthcare services is shortly explained in the Discussion section (lines 489 – 503). The theoretical framework should not pop up in the Discussion section but should be included in detail in the Introduction section. Thus, the authors should transfer the model's explanation to the manuscript's Introduction section.

3.      I would also propose thoroughly rethinking the terms used and defining them carefully. In the entire manuscript, different terms are not defined precisely like, e.g., unmet needs. If needs are unmet, the readers should be informed which needs are unmet and elaborate on the specific need structure and problems of the elderly in general as compared to the younger population. To understand why needs are unmet, it should be explained what kind of special needs exist, at least in my view. Why are chronic disease management, specialist care services, psychological distress, and formal caregiving listed under the theme "unmet need"? This is not reasonable to me. In my view, the specific needs of the elderly should be explained somewhere in the manuscript to refer to these needs later.

4.      The authors should include the verbatim interview guidelines for both settings, the interviews, and the group discussions. In the present form of the manuscript, it is unclear whether the two specific questions included on p. 2 (lines 76-78) were the research questions or whether these two questions were used in some respect also for the data collection. In the second question (line 78), "of older adults" should be included. Besides, a group discussion with only two participants is more or less a team discussion and not a group setting concerning its denomination.

5.      The authors should justify their decision to lead single interviews with older patients and group discussions with the health care professionals. I would expect it rather the other way round: single interviews with health care professionals and group discussions with patients. A group setting with other patients would lower the threshold for reporting unpleasant experiences and openly exchanging information in lively discussions. This threshold is much lower for professionals bearing their expertise in the field in mind. Thus, why did they opt for two different data collection procedures, and why did they not use the focus group setting for the patients and the interview setting for the health care professionals?

6.      In the Data Analysis section, the initial codes drawn out of Andersen's model of health services utilization should be explained in more detail.

7.      The authors should describe their samples and separate them according to the two data collection procedures. The characteristics and sociodemographic background of the participants should be delivered in the form of two Tables: one for the interviews and one for the group discussions. Were the group discussions homogeneous or heterogeneous in composition?

8.      In the Results section, the entire subsection on specialist care services deals with waiting time issues. However, later on, a particular theme was mentioned as "waiting time for appointments." Maybe both sections should be merged, or maybe the differentiation between the two sections should be made more comprehensible to the average readership.

9.      The entire section, "Suggested interventions to address access barriers," is not understandable. Are there any verbatim quotes to visualize what is meant thereby?

10.   The Discussion section is well-written in style and content. However, I would cordially invite the authors to underline what is new compared to what has been known before and refer to the existing literature in the field. Which categories and findings are new? These would constitute the core contribution of the manuscript. Thus, the authors should underline the new findings.

11.   The Conclusion section does not reveal conclusions but more or less takes the form of a summary of the results. The authors are cordially invited to be more innovative and to think about the implications for theory and practice. What can be done with the findings? What are the implications for different stakeholders within the healthcare system? What could be the effects on the elderly? Why are the findings important? The authors should elaborate on this.

Minor comments

-----------------------

12.   Some explanations on SA are given in the Research Design and Methods section. It might make sense to switch this part to the Introduction section. The authors should think about this suggestion.

13.   The authors should explain in more detail from which database they have drawn the e-mails to recruit the health professionals. Did the respondents receive any incentives?

14.   The entire Results section should be reported in the past tense. Currently, a mixture of present and past tense is included. Also, the formatting is inconsistent concerning the letters' size and using italics for verbatim quotes.

15.   A native speaker should read through the sentences again and correct all grammatical errors.

To sum up, I think the present manuscript under consideration could constitute a valuable contribution to the literature in the field. Nevertheless, I would invite the authors to address the issues raised and ensure that after the revision, the theoretical background should cohere with the deductive approaches taken for the analyses. Clear research aims should run like a golden thread through the whole manuscript.

Good luck with your research!

Author Response

Response document has been attached 

Round 2

Reviewer 2 Report

Dear authors!

Thank you for submitting a revised manuscript titled "Understanding unmet care needs of rural older adults with chronic health conditions: A qualitative study. " I appreciate the thorough revisions made, and, in my opinion, the quality of the manuscript has improved significantly through the first revision round. You have addressed nearly all of my comments satisfactorily and I have also read through the comments of the second reviewer, who raised similar concerns. Nonetheless, from my perspective, a few minor concerns should still be clarified before the manuscript can be published. So, I hope you will interpret my additional comments as constructive criticism, and I would be pleased if you smoothed them out, with a manageable effort in my view, to make your manuscript publishable soon. Altogether, I think the manuscript could make a nice work after these remaining minor points have been addressed. I will recommend its acceptance as far as the remaining issues have been addressed without the necessity of a further revision round. Thus, the editor should decide about a satisfactory addressing of the remaining two points of criticism.

Altogether, I appreciate the thorough revision with regard to several aspects. First of all, I appreciate the expansion of the content of the Introduction and the inclusion of Andersen's model of health services utilization. Thus, the structure of the Introduction and the manuscript, in general, is much clearer to me and hopefully for the average readership of IJERPH now.  

In terms of methodology, you have clarified reasonably, why you chose expert interviews for the older adults and described the research setting much more in detail, as was proposed also by the other Reviewer. It is now much more comprehensible what you have done and which tasks were taken over by whom of your research team. As it was proposed, the exact item wording of the interview guideline should be included in an appendix. The exact wording of the questions and the Introduction to the respondents are now appropriately disclosed. The manuscript has also profited very much from another round of proofreading it was undergone once again.

As I have already mentioned, the general topic is interesting and important. Thus, the subject of the available paper is up-to-date and, in my view, fascinating for the readership of the International Journal of Environmental Research and Public Health.

So, I would be pleased if you smoothed the remaining issues out quickly to make your manuscript ready for publication. Altogether, I think the manuscript will be publishable after addressing these remaining points. I want to elaborate on these points of criticism in detail as follows.

Remaining concerns

===============

1. In the first review round, I mentioned the following (Comment no. 7): "The authors should describe their samples and separate them according to the two data collection procedures. The characteristics and sociodemographic background of the participants should be delivered in the form of two Tables: one for the interviews and one for the group discussions. Were the group discussions homogeneous or heterogeneous in composition?"

You write in your authors' response that a table including the participants' characteristics and the sociodemographic background will not provide any new information apart from what has already been provided. This is not convincing, and I'm afraid I have to disagree with this. It is convenient to deliver some background information on the respondents, as can be seen when looking at papers that have also used a qualitative data collection approach and have been published recently in the IJERPH (e.g., Kim and Kim, 2021; Vincenzo et al., 2022). For the healthcare personnel, it is not even reported how many females and males were included. When quoting respondents, it is common practice to include an identifying albeit anonymous denomination reflecting the gender and age of the patients and the healthcare personnel, e.g., "P1, female, 64 yrs." If you did not collect the exact age, you could refer to age ranges. Thus, it is usual to include a summary of the respondents' descriptions, as seen in the paper by Vincenzo et al. (2022), and include a de-individualized, anonymous assignment of the respondent to each quote in the manuscript.

2. In the first review round, I mentioned the following (Comment no. 8): "In the Results section, the entire subsection on specialist care services deals with waiting time issues. However, later on, a particular theme was mentioned as "waiting time for appointments." Maybe both sections should be merged, or maybe the differentiation between the two sections should be made more comprehensible to the average readership." Your response is not convincing, and this issue has not been addressed properly.

All other points of criticism have been appropriately addressed. If you can smooth out the above-mentioned remaining leverage points of improvement, your manuscript could make a nice contribution to the literature. As soon as you have addressed the above-listed residual two points, I will recommend the publication of your manuscript.

References:

Kim, M., & Kim, M. (2021). We want more than life-sustaining treatment during end-of-life care: focus-group interviews. International Journal of Environmental Research and Public Health, 18(9), 4415.

Vincenzo, J. L., Patton, S. K., Lefler, L. L., McElfish, P. A., Wei, J., & Curran, G. M. (2022). Older Adults' Perceptions and Recommendations Regarding a Falls Prevention Self-Management Plan Template Based on the Health Belief Model: A Mixed-Methods Study. International Journal of Environmental Research and Public Health, 19(4), 1938.

Author Response

Response to reviewers' comments is attached 
